# Effects of Some Food Components on Non-Alcoholic Fatty Liver Disease Severity: Results from a Cross-Sectional Study

**DOI:** 10.3390/nu11112744

**Published:** 2019-11-12

**Authors:** Antonella Mirizzi, Isabella Franco, Carla Maria Leone, Caterina Bonfiglio, Raffaele Cozzolongo, Maria Notarnicola, Vito Giannuzzi, Valeria Tutino, Valentina De Nunzio, Irene Bruno, Claudia Buongiorno, Angelo Campanella, Valentina Deflorio, Annamaria Pascale, Filippo Procino, Paolo Sorino, Alberto Rubén Osella

**Affiliations:** 1Laboratory of Epidemiology and Biostatistics, National Institute of Gastroenterology, “S. de Bellis” Research Hospital, Castellana Grotte (Ba), Via Turi 27, 70013 Castellana Grotte, Italy; antonella.mirizzi@irccsdebellis.it (A.M.); isabella.franco@irccsdebellis.it (I.F.); carlaleone@interfree.it (C.M.L.); catia.bonfiglio@irccsdebellis.it (C.B.); irenebrunodiet@gmail.com (I.B.); buongiorno.claudia@gmail.com (C.B.); angelocampanella7@gmail.com (A.C.); valentinadeflorio@yahoo.it (V.D.); annamariapsc@hotmail.com (A.P.); filippo.procino@irccsdebellis.it (F.P.); paolosorino96@libero.it (P.S.); 2Clinic Gastroenterologic Unit, National Institute of Gastroenterology, “S. de Bellis” Research Hospital, Castellana Grotte (Ba), Via Turi 27, 70013 Castellana Grotte, Italy; raffaele.cozzolongo@irccsdebellis.it (R.C.); vito.giannuzzi@irccsdebellis.it (V.G.); 3Laboratory of Nutritional Biochemistry, National Institute of Gastroenterology, “S. de Bellis” Research Hospital, Castellana Grotte (Ba), Via Turi 27, 70013 Castellana Grotte, Italy; maria.notarnicola@irccsdebellis.it (M.N.); valeria.tutino@irccsdebellis.it (V.T.); valentinadx@hotmail.it (V.D.N.)

**Keywords:** NAFLD severity, foods groups components, Food Frequency Questionnaire

## Abstract

*Background:* The high prevalence of non-alcoholic fatty liver disease (NAFLD) observed in Western countries is due to the concurrent epidemics of overweight/obesity and associated metabolic complications, both recognized risk factors. A Western dietary pattern has been associated with weight gain and obesity, and more recently with NAFLD. *Methods:* This is a baseline cross-sectional analysis of 136 subjects (79 males) enrolled consecutively in the NUTRIATT (NUTRItion and Ac-TiviTy) study. Study subjects had moderate or severe NAFLD diagnosed by using Fibroscan-CAP. Food Frequency Questionnaire was used to obtain information about food intake. Statistical analysis included descriptive statistics and a multivariable logistic regression model. *Results:* The mean age was 49.58 (±10.18) with a mean BMI of 33.41 (±4.74). A significant inverse relationship was revealed between winter ice-cream intake and NAFLD severity (O.R. 0.65, 95% C.I. 0.95–0.99); chickpeas intake and NAFLD severity (O.R. 0.57, 95% C.I. 0.34–0.97), and not industrial aged-cheeses type (O.R. 0.85, 95% C.I. 0.74–0.98). A statistically significant positive association also emerged between rabbit meat (O.R. 1.23, 95% C.I. 1.01–1.49), industrial type aged cheeses (O.R. 1.17, 95% C.I. 1.01–1.35), milk-based desserts (no winter ice cream) (O.R. 1.11, 95% C.I. 1.01–1.21), fats (O.R. 1.12, 95% C.I. 1.01–1.25), and NAFLD severity. *Conclusion:* The fresh foods from non-intensive farming and high legume intake that characterize the Mediterranean diet would seem to be beneficial for patients with NAFLD.

## 1. Introduction

Non-alcoholic fatty liver disease (NAFLD), characterized by excessive hepatic fat accumulation, is the most common liver disorder in Western countries and affects 17–46% of adults [1]. Studies conducted in the populations of Castellana Grotte and Putignano (Apulian Region, Italy) have confirmed that NAFLD is also highly prevalent (about 30%) in this area, especially among elderly and male individuals [2,3]. Lifestyle interventions are the only known effective treatment for NAFLD [4,5], and changes in the dietary composition and consumption of individual food groups may have beneficial effects on NAFLD [6]. The way the food is produced, and how the animals are fed has been suggested to play a crucial role in health. Long-term bioaccumulation of antibiotic residues, for instance, may result in bacterial resistance [7,8], and gastrointestinal and liver problems [9]. The NUTRIATT (NUTRItion and Ac-TiviTy) Study is a nutritional randomized clinical trial (RCT), which enrolled subjects with moderate or severe NAFLD; it was a parallel-group randomized clinical trial aimed at estimating, as primary outcomes, the effect on NAFLD severity and the lipid red blood cells membrane profile of two different physical activity programs, of a Low Glycemic Index Mediterranean Diet (LGIMD) and their interaction as compared with a control diet. In this paper, we used baseline data to estimate associations between the consumption of some food group components with the grade of severity in NAFLD subjects enrolled in the NUTRIATT study before the intervention, while controlling for other food groups and demographics and biochemical characteristics.

## 2. Materials and Methods 

### 2.1. Study Design

Details of the study design have been published elsewhere [10]. Briefly, NUTRIATT was designed as a randomized clinical trial (RCT) (https://www.clinicaltrials.gov, NCT02347696) which enrolled subjects with NAFLD in both hospital and general practitioners’ settings.: Inclusion criteria were: (1) Body mass index (BMI) ≥25 kg/m^2^; (2) age >30 years old <60; (3) NAFLD, moderate or severe. Exclusion criteria included: (1) Overt cardiovascular disease and revascularization procedures; (2) stroke; (3) clinical peripheral artery disease; (4) current treatment with insulin or oral hypoglycemic drugs; (5) fasting glucose >126 mg/dL, or casual glucose >200 mg/dL; (6) more than 20 g/day of alcohol intake; (7) medical conditions that may impair the participation in a nutritional intervention study; (8) people following a special diet, involved in a weight loss program, who had experienced recent weight loss, and inability to follow a diet for religious or other reasons. The study was conducted from March 2015 to December 2016. Subjects were encouraged to participate in the study by their general practitioners, and then they were followed during the study by Nutritionists regarding adherence to the Mediterranean Diet and by Motor Sciences graduates for the adherence to the physical activity program. Subjects received personal counseling from the initial phase of the study in order to complete all kinds of documentation. In this study, we present data from the NUTRIATT study obtained at baseline. All subjects gave their informed consent to participate, and the study was conducted according to the Declaration of Helsinki and approved by the Ethics Committee (Prot. n. 10/CE/De Bellis, 3 February 2015).

### 2.2. Sample Size 

The trial sample size was estimated considering the repeated measurement nature of the outcome. From a previous study [11], the mean (±SD) score of NAFLD was estimated to be 4.5 (SD 1) and 4.0 (SD 0.5), corresponding to 251–299 dB and ≥300 dB on the vibration-controlled elastography scale [12,13], for the treatment and control group, respectively; the type I probabilistic error was fixed at 0.05 (one-sided) and statistical power at 0.9 (type II probabilistic error 0.10). The correlation between baseline/follow-up measurements of the outcome was set at 0.4. A sample size of *n1* = *n2* = *n3* = *n4* = *n5* = *n6* = 20 was estimated to obtain a lower categorization of NAFLD after three months. 

As this is a cross-sectional study we further investigated the minimum effect size detectable with different combinations of type I and II probabilistic errors for a sample size of 136 allocated in 2 groups. The maximum delta obtained was 0.4388 with a power of 95% and a type I probabilistic error of 0.001. All differences among groups for different foods in our sample size were in this range.

### 2.3. Data Collection

At enrollment, patients were interviewed by trained nutritionists to collect data about medical history and lifestyle. Anthropometric (waist circumference (WC), hip circumference (HC), height, weight) and bio-impedentiometric analyses (Akern SRL, Via Lisbona 32/34, 50065 Pontassieve (FI) Italy) were also performed. A blood sample was drawn after 12 h fasting and frozen at −80 °C until examination. All the analyses were performed within three months. 

Pre-coded questionnaires were used to collect information on alcohol consumption, smoking, education (number of years) and the occurrence of myocardial infarction, stroke, diabetes. Trained interviewers collected information. The assessment of alcohol intake was performed using a questionnaire [14], and a personal interview.

Information regarding physical activity and the number of hours of sleep was collected using the International Physical Activity Questionnaire (IPAQ) [15].

Trained dieticians collected anthropometric data at the first visit. WC was taken using a measuring tape while the subject was standing opposite the nutritionist, wearing underwear, feet joined, abdominal muscles relaxed, arms hanging down the body. Weight was taken while the subject was standing, wearing underwear, on an electronic balance, SECA^®^. Weight was approximated to the nearest 0.1 kg. Height was measured with a wall-mounted stadiometer SECA^®^, approximated to 1 cm, while the subject was standing barefoot, with the heels joined at an angle of 60°, the head on the Frankfurt horizontal plane parallel to the floor, arms hanging down the body with the palms facing the legs, and the shoulders, and buttocks back against the wall; the measure was performed after a deep inspiration, at the highest point. Blood pressure (BP) measurement was performed following international guidelines [16]. The average of 3 BP measurements was calculated.

### 2.4. Exposure Assessment

#### Foods Groups

The European Prospective Investigation into Cancer and Nutrition (EPIC) Food Frequency Questionnaire (FFQ) was used to obtain information about food intake [17]. The input of the EPIC FFQ was made online on the dedicated site for the Italian EPIC study [17], and centralized elaboration was carried out by the National Cancer Institute, based in Milan.

FFQs items were fitted into 21 major food groups. Major foods groups were chosen not only to provide a comprehensive representation of the local diet but also taking into consideration the glycemic index and nutritional composition of foods. These groups were: Milk-Yogurt, Sweet-Milk Based Foods, Aged Cheeses, Fresh Cheeses, Meats-Eggs, Meat products, Non-starchy Vegetables, Fruits, Dried Fruits, Refined-Grains, Whole-Grains, Starchy Vegetables, Legumes, Added Sugar Sweets, Pastry-Biscuit-Bread, Fats, Alcoholic Beverages, Tea-Coffee, Fish, Non-Alcoholic Beverages, and Sauces or Dressings.

### 2.5. Outcome Assessment

The diagnosis of NAFLD was performed using vibration-controlled elastography implemented on a FibroScan^®^ (Echosens, Paris, France). [18] NAFLD was categorized as absent (<215 dB), mild (215–250 dB), moderate (251–299 dB) and severe (≥300 dB) [12,13].

### 2.6. Statistical Methods

For descriptive purposes some variables were categorized: Age (30–40, 41–50, 51–60, 61 or >), Education (≤8 years, ≥9 years), Physical Activity (Low ≤ 4 Metabolic Equivalent of Task (MET)-minutes-week, Moderate = 4–5.99 MET-minutes-week, Vigorous ≥6 MET-minutes/week), Smoking (Never, Former, Current). BMI was categorized following World Health Organization standards, whereas WC cut-offs were <102 cm for men and <88 cm for women [19].

All data are expressed as mean (±SD) or percentages. Student’s t-test was performed to estimate differences in the mean intake of Food Groups between Moderate and Severe NAFLD subjects.

Multivariable Logistic Regression (MLR) models were then performed to estimate the association between the exposure variables and the outcome. In this analysis Age was introduced as continuous variables. Results from MLR are expressed as odds ratio (OR) and 95% confidence interval (95% CI) adjusted for demographic and all other food groups. To disentangle the effects of some food groups components on NAFLD severity, the following strategy was used. First, an MLR with Foods Groups as independent variables was fitted; then, MLR with the component of each Food Group separately to aisle single food group components effect, and successively an MLR with disease promoting or preventing foods components identified in the precedent analysis. In the final analysis, estimates were adjusted by food groups (without the components identified as promoters or preventing), sex, age and kcal. As each food group component was present in only one group collinearity does not matter. A probabilistic type I error of 0.05 was considered as statistically significant. Statistical analysis was performed using statistical software Stata (Version 15.1), StataCorp, 4905 Lakeway Drive, College Station, Lakeway, TX, USA.

## 3. Results

In total, 166 subjects were assessed for eligibility. Seventeen did not meet the inclusion criteria; 6 declined to participate and 7 had incomplete FFQ < 90% answers. Thus, 136 participants (58% males) were finally included in this study. Bio-sociocultural characteristics, nutritional status, and other variables of interest are shown in Table 1. Mean age (SD) of participants was 49.58 (10.18) years, and mean energy intake was 2315.90 (791.44) kcal. About 66% of subjects had higher education (university or tertiary education, completed or not). More subjects with overweight and different grades of obesity belonged to the Severe NAFLD category as well as higher waist circumference both in men and women. Severe NAFLD subjects had lower levels of physical activity and were current smokers. Seventy-eight percent of Moderate NAFLD subjects had F0/F1 grade of Stiffness. There was a good agreement between Fatty Liver Index and Fibroscan diagnosis.

Table 2 shows Biochemical Characteristics in subjects with NAFLD of Nutriatt Study. Homeostatic model assessment of Insulin Resistance (HOMA-IR), Glucose and Hemoglobin A1c IFCC (International Federation of Clinical Chemistry and Laboratory Medicine unit of measure) were substantially higher in Severe NAFLD subjects as well as glutamate-oxalacetate transaminase (GOT) and glutamate-pyruvate transaminase (GPT), C-Peptide and Insulin.

Table 3 shows Odds Ratio (ORs) for Foods Groups. No statistically significant association was observed.

Table 4 shows fully adjusted ORs for food groups components, classified as disease protective or promoting foods in relation to NAFLD severity, respectively. It is worthy of note that the average daily consumption (g/day) of winter ice-cream was higher in moderate than severe NAFLD. A statistically significant negative association between NAFLD severity and winter ice-cream intake, chickpeas intake, and aged cheeses (local), was evidenced. Conversely, a statistically significant positive association was observed between NAFLD severity and white bread, fats, and rabbit meat.

## 4. Discussion

This study showed that some food groups components were associated with a lower or a higher risk of developing severe NAFLD, and that, within the same Food Group, some components with a protective or promoter action are present. In particular, winter ice-cream was associated with a 35% lower risk of severe NAFLD per g/day intake. Conversely, the Sweet Products Milk-Based food group (without winter ice-cream) resulted associated with an 11% higher risk of severe NAFLD per g/day intake. Among the Aged Cheeses food group, only those industrially produced were associated with a 17% higher risk of severe NAFLD per g/day intake.

Moreover, in the group of protective foods, in addition to winter ice-cream, chickpeas were associated with a 43% lower risk of severe NAFLD per g/day intake. Conversely for “promoting foods” in addition to industrially produced Aged cheeses, Sweet Products Milk-Based Food Group (without winter ice-cream), the Fats group comprising: seed oil, peanut oil, butter, sunflower oil, corn oil, margarine, olive oil, extra virgin olive oil, soy oil, were associated with a 12% higher risk of severe NAFLD per g/day intake. Also, rabbit meat was associated with a 23% higher risk of severe NAFLD per g/day intake. 

As regards olive oil, a component of the Mediterranean Diet well known for its healthy effects in human beings, we did not find any effect when it was isolated from the Fats Group. This finding is not surprising because a high olive oil consumption is widespread in this region and so all subjects are exposed to much the same level of intake [20].

### 4.1. Effect of Food Groups Components on NAFLD

#### 4.1.1. Legumes 

A higher intake of legumes, especially beans and lentils, has been associated with a lower waist circumference, serum cholesterol, and blood pressure [21] and lower prevalence of NAFLD [22]. Obesity is found in 30% to 100% of subjects with NAFLD, and steatosis is 4.6-fold higher in obese compared to normal-weight individuals [23]. Dietary protein may reduce weight by increasing energy consumption and stimulating satiety [24]. Moreover, fiber and phytate reduce the digestibility, energy availability, and glycemic response, leading to the satiety sensation [25]. A large proportion of NAFLD patients has type 2 diabetes [26], and previous studies have shown that the intake of nutrient-dense and low energy foods is inversely related to the type 2 diabetes risk [27]. The low glycemic index of legumes can have beneficial effects on type 2 diabetes [28]. Moreover, it has been shown that the consumption of pulses improves glycemic control markers in individuals with or without type 2 diabetes [29], and could improve the lipid profile and decrease lipid peroxidation due to their high fiber content and low glycemic index [30].

In this region, pulses intake is particularly high. “La Farinella di Putignano” is a flour obtained from the grinding of roasted chickpeas and barley with added salt. This product was prepared locally as from ancient times and is commonly used across all social classes [31]. It is used as a real seasoning, ending up with pasta in sauce, soups, vegetable salads, seasoned olives, or fresh figs and fruit. This precious food is still present today on the every-day table of Apulians. 

#### 4.1.2. Aged Cheeses

Cheese and milk products are a rich source of so-called bioactive peptides [32]. In particular, animal studies suggest that the renin-angiotensin-aldosterone system (RAAS) could be of significant importance in the pathogenesis of NAFLD. The angiotensin-converting enzyme inhibitors (ACE-I) and angiotensin receptor blockers (ARBs) are therefore a potentially useful therapeutic approach [33]. The renin-angiotensin system (RAAS) seems to play an important role also in hepatic inflammation and fibrogenesis. Angiotensin II (ANG II) is recognized to induce hepatic inflammation and to stimulate a range of fibrogenic actions predominantly throughout the AT1 receptor [34]. Aged Cheeses such as Parmesan and Grana Padano are characterized by high nutritional quality effects and contain substances that have a particular biological activity, and therefore they can be fully considered functional foods, according to the European Union guidelines [35].

To date, there would appear to be no studies linking the consumption of aged cheeses and NAFLD; however, the presence of peptides with an ACE-I action has been shown in aged cheeses like Parmesan/GP, and also an effect of ACE-I on preventing fatty liver in rats has been demonstrated. [33,34].

#### 4.1.3. Winter Ice-Cream

In winter, ice-cream intake is frequent, but exclusively craftsman-made not packaged. Non-industrial production of ice-cream is made with a different type of milk as the animals are fed in a different way, and so the milk has a different nutritional value.

The quantity and quality of milk produced by cows depend strongly on many factors. In this sense, the transition period is a critical moment in dairy cow-feeding in traditional systems, where cows are converted from pasture to an indoor system. The consequent changes in nutrition might affect the quantity and quality of the milk [36]. Conventional milk has a greater range of different parameter seasonal variations than organic milk. These differences are particularly evident in spring and autumn, during the indoor period, due to the different diet given in organic versus conventional farms [37].

The safety and quality of dairy products are considered to be of significant importance for human health [38]. Worldwide, about 50% of all antibiotics production is used in animal and agriculture applications [7,8]. Since the 1990s, the use of antibiotics has been regulated, and maximum residue limits (MRLs) have been set [39]. Nevertheless, residues are still reported to be present in milk, at levels exceeding the MRLs worldwide [9,38]. The risk of consuming milk containing antimicrobial residues, even when present below the MRL, is also of great concern in regard to human health. The long-term bioaccumulation of antibiotic residues may result in bacterial resistance [7,8], hypersensitivity reactions [7,39], gastrointestinal and liver problems [9], as well as triggering cancer, mutagenicity, and toxicity in humans [40].

#### 4.1.4. Rabbit Meat

Worldwide, rabbit meat is valued for its high nutritional properties [41]. Compared to other meat types (chicken, beef, and pork), it was found that rabbit meat was richer in calcium (21.4 mg/100 g) and phosphorus (347 mg/100 g) and lower in fat (9.2 g/100 g) and cholesterol (56.4 mg/100 g) [42]. The intensification of production systems and concurrent exposure to different stressors, antibiotics at sub-therapeutic levels, have widely affected rabbits. During the last decades, considerable amounts of antibiotics have been used in animal production. Therapeutic use of antibiotics is typically a high dose-short term regimen. Growth promoting use is typically the opposite, i.e., a low dose-long term administration, usually given in feed [43].

### 4.2. Antibiotics 

It has been proposed that gut microbiota may be involved in the pathogenesis of obesity, possibly through effects on energy balance, nutrient absorption, inflammatory pathways, and the gut–brain axis [44,45]. Human and animal studies have established a strong association between a perturbed microbiome and the development of obesity and related metabolic dysfunctions [45]. These studies showed that antibiotic exposure substantially altered the gut microbiota and that adding a high calorie, high-fat diet increased the effects of the altered microbiota on both adiposity and hepatic gene expression [46]. As the acquisition of the intestinal microbiota begins at birth, the disruption of the microbiota during maturation by low-dose antibiotic exposure can alter host metabolism and adiposity [46,47]. However, data linking gut dysbiosis with the severity of NAFLD remain poorly documented in humans. Only a few series with generally small sample sizes, heterogeneous populations (adult versus children), and different methods for gut microbiota evaluation (qPCR versus pyrosequencing) are available in the literature [48]. 

Some methodological issues need to be considered. As this is a RCT with strict inclusion criteria, the sample size calculation was performed to obtain precise estimates, and the outcome assessment method has high sensitivity and specificity. Transient elastography with controlled attenuation parameter has demonstrated good accuracy in quantifying not only the levels of liver steatosis but also fibrosis in NAFLD. The method is fast, reliable and reproducible and has good intra- and interobserver levels of agreement. This characteristic gives it good properties to be used in population studies. Moreover, some initial technical difficulties to screen obese subjects have been overcome with the development of specific probes [49].

Reverse causation could matter in this type of patients. Reverse causation or temporal bias, ordinarily refers to the situation in which the outcome precedes and causes the exposure instead of the other way around [50]. It is unlikely that our subjects had changed their eating habits because in the majority of cases they were not aware of NAFLD. The subjects did not seek medical attention and came to participate as a way of improving their lifestyle, as indicated by the generalist.

Moreover, several confounding factors were considered to ensure that the estimates could be not only precise but also accurate. Limitations include the use of FFQ, although the instrument is highly standardized. In this case, exposure misclassification matters. However, if an exposure misclassification bias were introduced in the design, as the proportion of exposed/non-exposed subjects does not depend on the outcome, any bias introduced by nondifferential misclassification of the exposure is predictably only in one direction, namely, toward the null hypothesis [51]. 

## 5. Conclusions

In conclusion, there were consistent associations, either after adjustment, among some food group components and severity of NAFLD. These interesting associations need to be verified with longitudinal studies as it could help to elaborate personalized dietary counseling to treat NAFLD. It also emerged that the way food is produced, and the way animals are bred, would seem to play a role in rendering these foods promoters of the risk of worsening NAFLD.

## Figures and Tables

**Table 1 nutrients-11-02744-t001:** Bio-Sociocultural Characteristics, Nutritional status and other Variables of Participants. NutriAtt Study. Castellana Grotte (BA), Italy.

	NAFLD ^†^					
	Moderate		Severe		Total	
	No.	%	No.	%	No.	%
Age						
30–40	9	34.6	17	65.4	26	100.0
41–50	20	41.7	28	58.3	48	100.0
51–60	13	35.1	24	64.9	37	100.0
60+	3	12.0	22	88.0	25	100.0
Sex						
Men	26	32.9	53	67.1	79	100.0
Women	19	33.3	38	66.7	57	100.0
Education						
≤8 years	13	28.3	33	71.7	46	100.0
≥9 years	32	35.6	58	64.4	90	100.0
Body Mass Index						
25–29.9	13	41.9	18	58.1	31	100.0
30–34.9	27	42.2	37	57.8	64	100.0
35–39.9	3	12.0	22	88.0	25	100.0
≥40	2	12.5	14	87.5	16	100.0
Waist Circumference Women						
≤88 cm	8	80.0	2	20.0	10	100.0
>88 cm	11	23.4	36	76.6	47	100.0
Waist Circumference Men						
≤102 cm	16	47.1	18	52.9	34	100.0
>102 cm	10	22.2	35	77.8	45	100.0
Level of Physical Activity						
Low	2	22.2	7	77.8	9	100.0
Moderate	13	40.6	19	59.4	32	100.0
High	30	33.7	59	66.3	89	100.0
Smoking status						
Never smoked	29	36.3	51	63.7	80	100.0
Former smoker	8	25.0	24	75.0	32	100.0
Current smoker	8	33.3	16	66.7	24	100.0
Stiffness						
F0	9	20.0	12	14.0	21	100.0
F0/F1	26	58.0	47	55.0	73	100.0
F1	1	2.0	3	3.0	4	100.0
F2	6	13.0	9	10.0	15	100.0
F3	3	7.0	12	14.0	15	100.0
F4	0	0.0	3	3.0	3	100.0
Fatty Liver Index						
≤60%	23	51.0	21	23.0	46	100.0
>60%	22	49.0	70	77.0	92	100.0
Total	45	33.1	91	66.9	136	100.0

^†^ NAFLD: Non-alcoholic fatty liver disease. Due to missing date some cells does not add up to 136.

**Table 2 nutrients-11-02744-t002:** Biochemical Characteristics by NAFLD Severity. NutriAtt Study. Castellana Grotte (BA), Italy.

	NAFLD ^†^	
	Moderate	Severe
	Mean (SD)	Mean (SD)
SBP (mmHg) ^‡^	123.44 (18.43)	126.28 (14.21)
DBP (mmHg)	80.78 (9.35)	81.83 (7.28)
Homa-IR ^§^	2.06 (0.85)	3.43 (2.22)
Glucose (mg/dl)	92.20 (7.16)	101.51 (21.56)
Hemoglobin A_1C_ IFCC (mmol/mol)	37.44 (4.18)	40.48 (8.18)
Urea (mg %)	35.67 (9.54)	34.30 (9.23)
Creatinine (mg/dL)	0.78 (0.18)	0.79 (0.18)
eGFR (mL/min/1.73 mq) µ	108.59 (100.59)	85.98 (20.44)
Total Bilirubin (mg/dL)	0.66 (0.39)	0.60 (0.29)
Direct Bilirubin (mg/dL)	0.16 (0.05)	0.16 (0.05)
GOT (UI/L) ^¡^	23.36 (4.93)	26.85 (9.89)
GPT (UI/L) ^¥^	28.60 (10.17)	35.73 (20.29)
GGT (UI/L) ^!^	22.56 (10.84)	28.88 (22.73)
Iron (mcg%)	86.69 (32.01)	99.42 (100.34)
Total Cholesterol (mg %)	203.76 (35.66)	198.32 (40.47)
HDL Cholesterol (mg/dL) ^#^	49.67 (13.18)	42.38 (10.11)
Triglycerides (mg %)	108.18 (60.95)	143.05 (93.62)
Ceruloplasmin (mg/dL)	32.45 (9.19)	31.27 (8.05)
Alpha1 Antitrypsin (mg/dL)	134.51 (19.37)	137.40 (18.01)
FT3 (pg/mL) ^¿^	3.46 (0.47)	3.38 (0.37)
FT4 (ng/mL) ^$^	0.85 (0.10)	0.87 (0.13)
Cortisol (μg/dL)	11.87 (4.74)	10.01 (4.17)
C-peptide (ng/mL)	2.53 (0.70)	3.38 (1.31)
Insulin (μUI/mL)	9.02 (3.61)	13.59 (7.27)
Ferritin (ng/mL)	140.71 (146.19)	139.41 (152.69)
Folate (ng/mL)	6.24 (2.84)	6.01 (2.73)
Vit. B 12 (pg/mL)	308.73 (96.89)	321.18 (106.34)
KCAL	2385.80 (726.51)	2281.33 (823.33)

^†^ NAFLD: Non-alcoholic fatty liver disease; ^‡^ SBP: Systolic blood pressure; ^§^ DBP: Diastolic blood pressure; HOMA-IR. Homeostatic model assessment of Insulin Resistance; µeGFR. Estimated glomerular filtration rate; ^¡^ GOT glutamate-oxalacetate transaminase; ^¥^ GPT glutamate-pyruvate transaminase; ^!^ GGT gamma-glutamil transferase; ^#^ HDL cholesterol: High density lipoprotein cholesterol; ^¿^ FT3 free triiodothyronine; ^$^ FT4 Free thyroxine.

**Table 3 nutrients-11-02744-t003:** Food Groups by NAFLD Severity. NutriAtt Study. Castellana Grotte (BA), Italy.

	NAFLD ^†^		
	Moderate	Severe	
Foods (g/day)	Mean (a)	Mean (a)	OR (95% CI) (a)
Milk And Yogurt	450.21	448.79	0.99 (0.99–1.00)
Sweet Products Milk Based	17.75	18.37	1.00 (0.97–1.03)
Aged Cheeses	34.79	26.23	0.98 (0.97–1.00)
Cheeses	23.59	22.93	0.99 (0.98–1.02)
Meets And Eggs	117.78	133.60	1.00 (0.99–1.01)
Meat Products	20.64	20.12	0.99 (0.98–1.02)
Non-Starchy Vegetables	152.50	166.43	1.00 (0.99–1.01)
Fruits	383.33	314.78	0.99 (0.99–1.00)
Dried Fruits	11.30	9.92	0.99 (0.98–1.01)
Refined Grains	74.89	79.85	1.00 (0.99–1.01)
Whole Grains	6.80	6.32	0.99 (0.96–1.03)
Legumes	50.02	42.66	0.99 (0.98–1.00)
Starchy Vegetables	38.82	46.93	1.01 (0.99–1.02)
Added Sugar And Sweets	33.81	25.26	0.99 (0.97–1.00)
Paste, Bisquits And Bread Rolls	217.23	216.86	0.99 (0.99–1.00)
Fats	23.72	27.37	1.03(0.99–1.07)
Alcoholic Beverages	115.79	165.25	1.00 (0.99–1.00)
Tea Coffee	125.95	133.41	1.00 (0.99–1.00)
Non-Alcoholic Caloric Beverages	91.27	125.79	1.00 (0.99–1.00)
Souces Dressings	7.40	9.97	1.02 (0.99–1.05)
Fish	41.14	42.40	1.00 (0.99–1.02)

(a) Age, sex, and Kcal adjusted mean and ORs; ^†^ NAFLD: Non-alcoholic Liver Fatty Disease.

**Table 4 nutrients-11-02744-t004:** Protective and promoting foods by NAFLD severity. NutriAtt Study. Castellana Grotte (BA), Italy.

	NAFLD ^†^			
	Moderate	Severe		
Foods (g/day)	Mean (a)	Mean (a)	OR (95% CI) (a)	OR (95%CI) (a, b)
Protective Foods				
Chocolate	6.99	4.02	0.95 (0.90–0.99) **	0.94 (0.84–1.05)
Winter Icecream	2.21	0.88	0.82 (0.70–0.96) **	0.65 (0.47–0.89) *
Apricoats	8.35	4.94	0.95 (0.90–0.99) **	0.95 (0.82–1.10)
Pears	56.84	35.09	0.99 (0.99–0.99) **	0.99 (0.98–1.01)
Soya Milk	15.59	5.04	0.98 (0.97–0.99) **	0.99 (0.97–1.02)
Legumes-Rice	1.60	0.68	0.83 (0.69–0.98) **	0.73 (0.50–1.06)
Chickpeas	1.95	1.19	0.71 (0.56–0 92) *	0.57 (0.34–0.97) **
Dried Peas	1.63	0.97	0.69 (0.51–0.94) **	0.78 (0.44–1.39)
Local Aged Cheeses	34.79	26.23	0.99 (0.98–1.00)	0.85 (0.74–0.98) **
Promoting Foods				
Industrial Aged Cheeses	31.85	25.41	0.88 (0.78–0.99) **	1.17 (1.01–1.35) **
White Bread	17.81	39.08	1.01 (1.00–1.02) **	1.02 (0.99–1.04)
Sweet Milk-Nowinter Icecream	17.75	18.37	1.01 (0.98–1.05)	1.11 (1.01–1.21) **
French Fries	3.23	6.58	1.06 (0.99–1.12)	1.10 (0.99–1.24)
Fats	23.72	27.37	1.03 (0.99–1.07)	1.12 (1.01–1.25) **
Rabbit Meat	1.71	3.80	1.12 (1.01–1.24) **	1.23 (1.01–1.49) **

(a) Age, sex, and Kcal adjusted mean and ORs; (b) Adjusted for all food groups without protective foods and promoting foods that are present in the table. ^†^ NAFLD: Non-alcoholic fatty liver disease * *p* < 0.01; ** *p* < 0.05.

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
