# Peer review of "Effects of Some Food Components on Non-Alcoholic Fatty Liver Disease Severity: Results from a Cross-Sectional Study"

_nutrients, 2019, doi:10.3390/nu11112744_

Round 1
Reviewer 1 Report
Thank you for revising the manuscript. The revised one is now suitable for publication.
Reviewer 2 Report
In the revised version of the manuscript the authors have addressed the issues delineated in the first review. No further changes recommended on the second review.
This manuscript is a resubmission of an earlier submission. The following is a list of the peer review reports and author responses from that submission.
Round 1
Reviewer 1 Report
In this cross-sectional study, an association between food groups/components and severity of the fatty liver was investigated. The findings show that certain food groups (such as cheeses) may contain individual components with opposite outcome associations (aged cheeses, local vs. industrial). These are interesting findings.
However, there are some concerns related to the manuscript.
Methods/Statistical analysis
Statistical analyses are not described in detail, and there seems to be some controversial information available.
-It was described in the methods section (page 4, row 146) that biochemical characteristics (content of variables in detail?) were included in the MLR models. However, this is not mentioned in connection with the tables.
-Was it described in the methods section (page 4, row 141) that Student’s t-test was adjusted for age, sex and Kcal. How is this possible?
-There were 136 subjects in this study. It seems that fully adjusted MLR models (Table 4) contained a plenty of different covariates, maybe too many? Some of these covariates may also strongly correlate with each other. How are these things handled/justified?
-It was described that some variables, such as age, were categorized for statistical analyses (page 3, row 136). Does this categorization concern also the MLR models?
Results
Is energy intake data available? There might be sense to use energy-adjusted intakes since portion sizes are dependent on the amount of energy needed.
Discussion
Usability/reliability of the Fibroscan as a method to assess severity of the fatty liver should be shortly discussed.
Conclusions
Only associations were reported by this cross-sectional study. For this reason, there is no evidence regard to causality, or no evidence regard to certain food groups or components to prevent or accelerate the disease progression.
Minor comments:
Abstract is presented partly twice.
Author Response
We thank the reviewer and include here our point-by-point answers
Reviewer 1
In this cross-sectional study, an association between food groups/components and severity of the fatty liver was investigated. The findings show that certain food groups (such as cheeses) may contain individual components with opposite outcome associations (aged cheeses, local vs. industrial). These are interesting findings.
However, there are some concerns related to the manuscript.
Methods/Statistical analysis
Statistical analyses are not described in detail, and there seems to be some controversial information available.We have modified the statistical analysis section. Now, it reads: Multivariable Logistic Regression (MLR) models were then performed to estimate the association between the exposure variables and the outcome. In this analysis Age was introduced as continuous variables. Results from MLR are expressed as Odds Ratio (OR) and 95% Confidence Interval (95% CI) adjusted for demographic, biochemical characteristics, and all other food groups. To disentangle the effects of some food groups components on NAFLD severity, the following strategy was used. First, an MLR with Foods Groups as independent variables was fitted; then, MLR with the component of each Food Group separately to aisle single food group components effect, and successively an MLR with disease promoting or preventing foods components identified in the precedent analysis. In the final analysis, estimates were adjusted by Food Groups (without the components identified as promoters or preventing), sex, age and kcal. As each Food Group component was present in only one group collinearity does not matter. A probabilistic type I error of 0.05 was considered as statistically significant. Statistical analysis was performed using statistical software Stata (Version 15.1), StataCorp, 4905 Lakeway Drive, College Station, Texas 77845 USA.
It was described in the methods section (page 4, row 146) that biochemical characteristics (content of variables in detail?) were included in the MLR models. However, this is not mentioned in connection with the tables.
We have modified this paragraph as this variables were not considered in the analysis as these variables did not modify the precision and validity of estimates.
3) Was it described in the methods section (page 4, row 141) that Student’s t-test was adjusted for age, sex and Kcal. How is this possible?
It was a typo and now we have written the right sentence.
There were 136 subjects in this study. It seems that fully adjusted MLR models (Table 4) contained a plenty of different covariates, maybe too many? Some of these covariates may also strongly correlate with each other. How are these things handled/justified?
We have added a sentence in the statistical analysis section. Now it reads: To disentangle the effects of some food groups components on NAFLD severity, the following strategy was used. First, an MLR with Foods Groups as independent variables was fitted; then, MLR with the component of each Food Group separately to aisle single food group components effect, and successively an MLR with disease promoting or preventing foods components identified in the precedent analysis. In the final analysis, estimates were adjusted by Food Groups (without the components identified as promoters or preventing), sex, age and kcal. As each Food Group component was present in only one group collinearity does not matter.
5) It was described that some variables, such as age, were categorized for statistical analyses (page 3, row 136). Does this categorization concern also the MLR models?
We agree with the reviewer. This was confusing. We have changed the paragraph. It now reads: For descriptive purposes some variables were categorized: Age (30-40, 41-50, 51-60, 61or >), Education (≤8 years, ≥9 yr), Physical Activity (Low=< 4 Metabolic Equivalent Of Task (MET)-minutes-week, Moderate=4-5.99 MET-minutes-week, Vigorous=≥6 MET-minutes-week), Smoking (Never, Former, Current). BMI was categorized following World Health Organization standards, whereas WC cut-offs were <102 cm for men and < 88 cm for women [18].
And:
There was an error in the paragraph and now it reads:
Multivariable Logistic Regression (MLR) models were then performed to estimate the association between the exposure variables and the outcome. In this analysis Age was introduced as continuous variables.
Results
6) Is energy intake data available? There might be sense to use energy-adjusted intakes since portion sizes are dependent on the amount of energy needed.
Only total energy intake data were available and we did not split this quantity to estimate the energy intake corresponding to each food group or food group component, then we did not perform Willet’s regression to obtain residuals. There were technical problems and time consuming procedures to perform this type of analysis. Only Total energy intake was introduced in the model.
Discussion
7) Usability/reliability of the Fibroscan as a method to assess severity of the fatty liver should be shortly discussed.
We have added in the limitations and strengths section of Discussion a paragraph. It now reads: Transient elastography with controlled attenuation parameter has demonstrated good accuracy in quantifying not only the levels of liver steatosis but also fibrosis in NAFLD. The method is fast, reliable and reproducible and has good intra- and interobserver levels of agreement. This characteristic gives it good properties to be used in population studies. Moreover some initial technical difficulties to screen obese subjects have been overcome with the development of specific probes.(Reference)
Conclusions
8) Only associations were reported by this cross-sectional study. For this reason, there is no evidence regard to causality, or no evidence regard to certain food groups or components to prevent or accelerate the disease progression.
We agree with the reviewer. Conclusions have been changed. It now reads: In conclusion, there were consistent associations, either after adjustment, among some food group components and severity of NAFLD. These interesting associations need to be verified with longitudinal studies as it could help to elaborate personalized dietary counseling to treat NAFLD. It also emerged that the way food is produced, and the way animals are bred, would seem to play a role in rendering these foods promoters of the risk of worsening NAFLD.
Minor comments:
Abstract is presented partly twice.
The abstract partially present in the manuscript has been cancelled
Reviewer 2 Report
Mirizzi et al present data here from the NUTRIATT study specifically focusing on how certain foods effect fatty liver disease severity. The study as designed and presented does show associations between particular food intake and NAFLD severity but fails to show causality. Comments and recommendations are as follows:
Certain types of foods are described as being protective or promoting. This gives the connotation that the particular foods may be therapeutic or harmful. While the data show that there are associations between particular foods and disease severity, the study is not designed to prove that a particular food is causative or therapeutic in NAFLD progression. Also, this is a snapshot in time of what food types are being reported. Following whether the fibroscan score increased or decreased over a set time period would provide stronger evidence. No control group is included. The initial publication from the NUTRIATT study referenced has control patients. Were food intake questionnaires given to those individuals? The primary outcome utilized is fibroscan but no reference is provided for why the cutoffs decided on were utilized. The existing reference noted in the paper is a commentary on a study looking at the effectiveness of fibroscan relative to biopsy. I would at least recommend referencing the primary study. It would be more beneficial to reference support for the cutoffs chose as review of studies utilizing fibroscan as an endpoint show variable cutoffs. Only steatosis data is included. Typically fibroscan provides a stiffness index as an assesment of liver fibrosis. Was this data collected? The results section of the manuscript only provides tables with very little written commentary of the results. It would be very beneficial with reference to each individual table.Author Response
We thank the reviewer and include our answers here.
Mirizzi et al present data here from the NUTRIATT study specifically focusing on how certain foods effect fatty liver disease severity. The study as designed and presented does show associations between particular food intake and NAFLD severity but fails to show causality. Comments and recommendations are as follows:
Certain types of foods are described as being protective or promoting. This gives the connotation that the particular foods may be therapeutic or harmful.
While the data show that there are associations between particular foods and disease severity, the study is not designed to prove that a particular food is causative or therapeutic in NAFLD progression.
Also, this is a snapshot in time of what food types are being reported.
We have improved the objective of the paper and stressed in the discussion section the cross-sectional nature of the design. Now it reads:
In this paper, we used baseline data to estimate associations between the consumption of some food components with the grade of severity in NAFLD subjects enrolled in the NUTRIATT study before the intervention, while controlling for other food groups and demographics and biochemical characteristics.
And
In conclusion, there were consistent associations, either after adjustment, among some food group components and severity of NAFLD. These interesting associations need to be verified with longitudinal studies as it could help to elaborate personalized dietary counseling to treat NAFLD. It also emerged that the way food is produced, and the way animals are bred, would seem to play a role in rendering these foods promoters of the risk of worsening NAFLD.
Following whether the fibroscan score increased or decreased over a set time period would provide stronger evidence.
We agree with the reviewer and fibroscan was repeated two more times other than the enrollment. These fibroscan were performed while enrolled subjects underwent the intervention (diet, physical activity or both). However our objective in this paper was to probe association between NAFLD and diet before any intervention.
No control group is included. The initial publication from the NUTRIATT study referenced has control patients. Were food intake questionnaires given to those individuals?
The reviewer is right. There is a group control in the two publication from Nutriatt. However, for those pubblications we took people who did not satisfy the inclusion criteria as they had NAFLD absent or mild. Those subjects did not underwent the complete screening program (only a fast blood venous sample, fiborscan and anamnesis) as the principal criteria was absent, then they did not complete the FFQ so dietary data are not available.
The primary outcome utilized is fibroscan but no reference is provided for why the cutoffs decided on were utilized. The existing reference noted in the paper is a commentary on a study looking at the effectiveness of fibroscan relative to biopsy. I would at least recommend referencing the primary study. It would be more beneficial to reference support for the cutoffs chose as review of studies utilizing fibroscan as an endpoint show variable cutoffs
We have now included one new references about Fibroscan.
Only steatosis data is included. Typically fibroscan provides a stiffness index as an assesment of liver fibrosis. Was this data collected?
Data about stiffness have been now included in Table 1 and commented in the results section
The results section of the manuscript only provides tables with very little written commentary of the results. It would be very beneficial with reference to each individual table.
We have added more comments about data present in tables in the results section
Reviewer 3 Report
The authors studied the dietary habits of 136 patients with moderate or severe Non-alcoholic fatty liver disease (NAFLD) and concluded that consumption of some foods such as "winter ice-cream" or rabbit meat were associated with higher severity of NAFLD determined by fibroscan.
Since the results are truly surprising, this reviewer suggest the following in other to increase the scientific value of the manuscript and to confirm in a re-analyses the validity of the results:
The sample size seems too low (n=136). The number of participants to be included should be assessed by statistical methods. The diagnosis of NAFLD should be improved. Nowadays, no liver biopsy is needed in the diagnosis of this disease, but it should be based in blood tests (including elevated liver enzymes) and the results of an imaging technique such as ultasonography scan (or CT scan). Fibroscan is very useful to assess the stiffness of the liver, a measurement that correlates with the grade of fibrosis. A control group and/or patients with mild NAFLD should be included in they. The possibility of "reverse causality" should be discussed in deph.Author Response
We thank the reviewer and include our answers here.
The authors studied the dietary habits of 136 patients with moderate or severe Non-alcoholic fatty liver disease (NAFLD) and concluded that consumption of some foods such as "winter ice-cream" or rabbit meat were associated with higher severity of NAFLD determined by fibroscan.
Since the results are truly surprising, this reviewer suggest the following in other to increase the scientific value of the manuscript and to confirm in a re-analyses the validity of the results:
The sample size seems too low (n=136). The number of participants to be included should be assessed by statistical methods. Vedere il sample size
We have added a commentary about sample size and mean differences at the baseline. It now reads:
The trial sample size was estimated considering the repeated measurement nature of the outcome. From a previous study [11], the mean (±SD) score of NAFLD was estimated to be 4.5 (SD 1) and 4.0 (SD 0.5), corresponding to 251-299 dB and ≥ 300 dB on the vibration-controlled elastography scale [12], for the treatment and control group, respectively; the type I probabilistic error was fixed at 0.05 (one-sided) and statistical power at 0.9 (type II probabilistic error 0.10). The correlation between baseline/follow-up measurements of the outcome was set at 0.4. A sample size of n1=n2=n3=n4=n5=n6=20 was estimated to obtain a lower categorization of NAFLD after three months.
As this is a cross-sectional study we further investigated the minimum effect size detectable with different combinations of type I and II probabilistic errors for a sample size of XXX equally allocated among 3 groups. The maximum delta obtained was 0.4388 with a power of 95% and a type I probabilistic error of 0.001. All differences among groups for different foods in our sample size were in this range.
The diagnosis of NAFLD should be improved. Nowadays, no liver biopsy is needed in the diagnosis of this disease, but it should be based in blood tests (including elevated liver enzymes) and the results of an imaging technique such as ultasonography scan (or CT scan).
We have now added data about Fat Liver Index in Table 1 which substantially confirm fibroscan diagnosis of NAFLD.
Fibroscan is very useful to assess the stiffness of the liver, a measurement that correlates with the grade of fibrosis.
Results about stiffness have now been included in the Tables and commented in the results section
A control group and/or patients with mild NAFLD should be included in they.
We agree with the reviewer and fibroscan was repeated two more times other than the enrollment. These fibroscan were performed while enrolled subjects underwent the intervention (diet, physical activity or both). However our objective in this paper was to probe association between NAFLD and diet before any intervention.
The possibility of "reverse causality" should be discussed in deph.
Reverse causation could matter in this type of patients. Reverse causation or temporal bias, ordinarily refers to the situation in which the outcome precedes and causes the exposure instead of the other way around (9–11). It is unlikely that our subjects had changed their eating habits because in the majority of cases they were not aware of NAFLD. The subjects did not seek medical attention and came to participate as a way of improving their lifestyle, as indicated by the generalist